# Cancer-cell intrinsic gene expression signatures overcome intratumoural heterogeneity bias in colorectal cancer patient classification

Philip D. Dunne[1], Matthew Alderdice[1], Paul G. O'Reilly[1], Aideen C. Roddy[1], Amy M.B. McCorry[1], Susan Richman[2], Tim Maughan[3], Simon S. McDade[1], Patrick G. Johnston[1], Daniel B. Longley[1], Elaine Kay[4], Darragh G. McArt[1,*] & Mark Lawler[1,*]

Stromal-derived intratumoural heterogeneity (ITH) has been shown to undermine molecular stratification of patients into appropriate prognostic/predictive subgroups. Here, using several clinically relevant colorectal cancer (CRC) gene expression signatures, we assessed the susceptibility of these signatures to the confounding effects of ITH using gene expression microarray data obtained from multiple tumour regions of a cohort of 24 patients, including central tumour, the tumour invasive front and lymph node metastasis. Sample clustering alongside correlative assessment revealed variation in the ability of each signature to cluster samples according to patient-of-origin rather than region-of-origin within the multi-region dataset. Signatures focused on cancer-cell intrinsic gene expression were found to produce more clinically useful, patient-centred classifiers, as exemplified by the CRC intrinsic signature (CRIS), which robustly clustered samples by patient-of-origin rather than region-of-origin. These findings highlight the potential of cancer-cell intrinsic signatures to reliably stratify CRC patients by minimising the confounding effects of stromal-derived ITH.

[1] Centre for Cancer Research and Cell Biology, Queen's University Belfast, Belfast BT9 7AE, UK. [2] Department of Pathology and Tumour Biology, Leeds Institute of Cancer and Pathology, St James Hospital, Leeds LS9 7TF, UK. [3] CRUK/MRC Oxford Institute for Radiation Oncology, University of Oxford, Oxford OX3 7DQ, UK. [4] Department of Histopathology, Beaumont Hospital and Royal College of Surgeons in Ireland, Dublin, Ireland. * These authors contributed equally to this work. Correspondence and requests for materials should be addressed to P.D.D. (email:p.dunne@qub.ac.uk) or to M.L. (email: mark.lawler@qub.ac.uk).

The application of transcriptional gene signatures to stratify tumours into prognostic and predictive subtypes has evolved rapidly since the original landmark studies demonstrated the clinical utility of this approach in breast cancer[1,2]. While the number of published signatures continues to increase, the likelihood of individual signatures achieving clinical utility has remained very low, with some estimates putting this figure below 1% (ref. 3). The factors dictating the eventual success or failure of a biomarker-based gene expression signature can often be attributed to statistical confounders such as insufficient sample size in the initial training set employed for discovery, or inadequate independent data sets for subsequent validation[4]. While it is possible to control these statistical variables with the addition of greater sample numbers or more relevant data sets, it is more difficult to mitigate against the problems associated with gene expression variations between different regions of the tumour, arising due to changes in tumour cell purity and/or stromal/immune infiltration. The degree of gene expression changes associated with variation in tumour microenvironment (TME) content; that is, stromal-derived intratumoural heterogeneity (ITH), may even mask the relatively more subtle changes associated with genetic variability and heterogeneity within the tumour epithelium[5]. This variation in TME content at different regions of the tumour has long been recognized[6]; however, in the era of precision medicine, the ramifications of such microenvironmental alterations in confounding patient classification could be of profound significance.

Recently, an international colorectal cancer subtyping consortium, tasked with defining the transcriptional landscape of CRC, published their assignment of four consensus molecular subtypes in CRC (CMS1-4)[7]. The poor-prognostic CMS4 subtype has been found to be heavily reliant on gene expression originating from the stromal component, particularly fibroblasts, within the TME[8,9]. Importantly, we have demonstrated that changes in region-specific tumour biology, associated with variation in stromal content and therefore the underlying stromal gene expression, can lead to inaccurate CRC subtyping using CMS[10]. These findings suggest that even subtle changes in stromal content may also undermine prospective patient stratification and disease management decisions, as the region-of-origin of the extracted tumour tissue is very often outside of the control of the molecular profiling team. It is imperative that patient classification using transcriptional signatures remains unaffected by biological variables arising due to stromal-derived ITH, thereby allowing robust identification of patient subtype, regardless of the location from which the tumour tissue has been extracted.

By evaluating differentially expressed genes between the central tumour (CT) and invasive front (IF) regions of primary colorectal tumours, we previously demonstrated that stromal-derived ITH can be a major confounder of transcriptomics-based CMS patient subtyping[10]. We have now extended our analyses to include gene expression profiles generated from lymph nodes (LN) matched to samples obtained from the CT and invasive edge of the primary disease. Using this extended dataset, we assessed how the performance of clinically relevant gene signatures can vary due to ITH. We demonstrate for the first time that transcriptomic signatures based on cancer cell-intrinsic gene expression overcome the confounding effects of TME-related ITH and group samples by patient-of-origin rather than region-of-origin. These findings have important implications for the clinical application of transcriptomics-based patient classification approaches.

## Results

### Clustering of multi-regional samples from primary tumours.
We have previously demonstrated how stromal-derived gene expression could undermine patient classification when using a gene signature associated with the recently published CRC CMS[10]. Using the random-forest (RF) methodology of generating classification scores for CMS subtypes (as defined in the original CMS study[7]) we now observe an increased relative CMS4 (mesenchymal subtype) classification score in IF samples when compared to their patient-matched CT samples for almost 90% of patients, which, in line with our previous work, we attribute to stromal-derived ITH (Fig. 1a). This increase is not observed in classification scores for CMS1 (microsatellite instability immune subtype) or CMS3 (canonical subtype), while the epithelial-rich CMS2 (metabolic subtype) displayed a general decrease in classification score in IF samples compared to the CT (Fig. 1a).

To investigate the extent to which stromal ITH can undermine the prediction of patient prognosis in CRC, we assessed the ability of four clinically relevant gene signatures, (namely Jorissen et al.[11], Eschrich et al.[12], Kennedy et al.[13] and Popovici et al.[14], see Methods section for detailed description of these signatures), to cluster the transcription profiles from patient-matched central tumour (CT, $n = 24$) and invasive front (IF, $n = 24$) (Fig. 1b). To include a suitable CMS signature for assessment in every analysis throughout our study, we firstly provide a clear demonstration of the utility of the Sadanandam et al.[20] CRC-assigner (CRCA) gene signature as a surrogate for the random-forest CMS classification system (Supplementary Fig. 1a). Using this approach, we observe 85–90% concordance in patient classification observed between CRCA and CMS subtypes in the GSE14333 CRC dataset (Supplementary Fig. 1b). In addition to the described signatures, we also include the specific stem-like CRCA classifier as a surrogate for CMS4 specifically, which we have previously proposed to be the classification subtype that is most prone to variation due to fibroblast content[10]. We confirm that over 97% of tumours from the GSE14333 data set classified as stem-like by CRCA were subsequently classified as CMS4, further validating this approach (Supplementary Fig. 1c). In addition, as a positive control for confounding variations in stromal-derived gene expression, we employed our previously published 30 gene signature, primarily fibroblast in origin and generated using differential expression between the CT and IF samples in this cohort, to stratify samples based on region-of-origin, regardless of patient-of-origin[10]. The gene sets we have selected did not undergo any additional adjustment or weighting during our analyses. Using the top-down divisive clustering analysis (DIANA) method, we observed 0% concordant clustering of our samples by patient-of-origin following semi-supervised clustering with this 30 gene signature (Patients labelled A-Y, Fig. 1c,d). The stem-like (CMS4) classifier clustered 21% of patients concordantly, further supporting the findings from our previous study. The CRC prognostic subtyping signatures generated by Jorissen et al.[11] (29%), Eschrich et al.[12] (38%) were poor at clustering samples according to patient-of-origin, while the CMS surrogate from Sadanandam et al.[20] (54%) displayed intermediate clustering. In contrast, the prognostic subtyping signatures of both Kennedy et al.[13] (88%) and Popovici et al.[14] (88%) demonstrated a profound increase in clustering samples based on patient-of-origin (Fig. 1c,d).

### Stability of patient classification across tumoural regions.
To further test the ability of each signature to robustly classify samples on a 'per patient' basis, regardless of the region-of-origin of the tissue sampled, we extended the multi-region dataset analysis to include gene expression data obtained from a matched lymph node metastasis for each patient (LN; $n = 24$, total dataset $n = 72$). Utilizing this cohort, we then employed a semi-supervised analysis of each signature, using hierarchical clustering of region-specific samples from both the primary tumour CT and IF, alongside the LN metastasis samples.

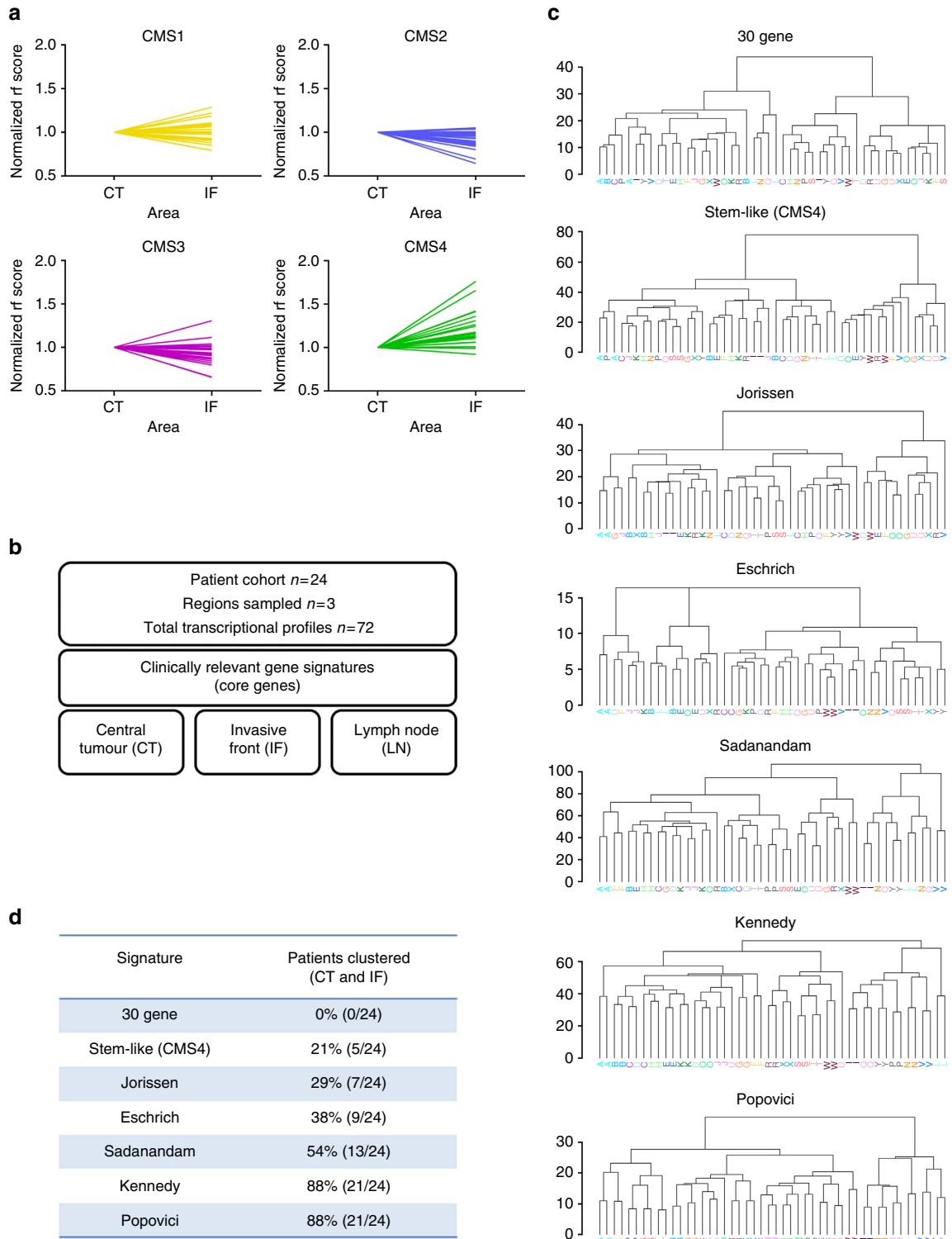

**Figure 1 | Variation in the ability of gene expression signatures to concordantly cluster multi-region samples according to patient-of-origin.**
(**a**) Random Forest (RF) classifier scores specifically for CMS1-4 individually in the patient-matched samples. RF scores for each patient were normalized to the CT sample (CT = 1 for all patients) and IF scores were plotted relative to this. Patients are labelled alphabetically (A–Y) and colour coded according to each individual CMS analysis for visualization (Yellow = CMS1, Blue = CMS2, Pink = CMS3, Green = CMS4). (**b**) Overview of the multi-region samples used in the analysis. Detailed information on each signature is outlined in the Methods section. Briefly, the 30 gene signature was developed as a classifier of region-of-origin in this dataset and can stratify samples into CT or IF regional groups. The Sadanandam signature is a surrogate marker of the CMS classifier and the stem-like signature is a sub-classifier within the Sadanandam signature specifically for the CMS4 subtype. The Jorissen, Eschrich and Kennedy signatures are stage II/III prognostic CRC classifiers. The Popovici signature classifies stage II/III CRC according to similarity to a BRAF mutant transcriptional classifier. (**c**) Divisive clustering methodology (DIANA) highlights the potential of each individual gene expression signature to correctly cluster multi-region primary tumour samples according to the patient-of-origin. Patients are labelled alphabetically (A–Y) and colour coded for visualization. (**d**) Table of concordantly clustered patient samples according to each signature. CT, central tumour; IF, invasive front.

This analysis confirmed the ability of our previously published stromal-associated 30 gene signature to identify regions-of-origin rather than patient-of-origin, with similar results observed for the stem-like CMS4 classifier (Fig. 2a,b). Wide variations in the ability of the remaining five published gene signatures to cluster transcriptional profiles by patient-of-origin were observed. Similar to the results of the initial DIANA analyses, we observed lower patient clustering for the Jorissen, Eschrich and Sadanandam signatures, when compared to either the Kennedy or Popovici signatures (Fig. 2c–g). On closer examination, we found that stratification of patient-matched samples was observed not only into different individual patient clusters, but also into distinct and opposing prognostic tumour subtypes (Fig. 2c–g). This finding suggests that classifiers based on genes present within the Jorissen, Eschrich or Sadanandam signatures could potentially misclassify patients based on the tissue region-of-origin, whereas those using genes represented in the Kennedy or Popovici signature would provide a more robust representation of tumour-specific signatures, not confounded by stromal ITH. Given that the proposed clinical utility of these signatures relates to their prognostic/predictive ability to guide disease management decisions, these initial findings suggest that the confounding ITH issues identified by ourselves and others[8–10] could undermine transcriptomics-based precision medicine-focused clinical interventions.

**Cancer-cell specific intrinsic gene expression**. To further assess the similarity of the multi-region samples for each patient, all seven gene expression signatures were tested using a non-clustering statistic (Pearson correlation coefficient analysis). To allow a quantitative comparison of both the intra- and inter-patient similarities for each signature, we implemented an additional normalization step in this analysis (detailed in the Pearson similarity section of the Methods section), by assessing the correlation between samples specifically from the same patient, compared to samples from different patients (Fig. 3a). Using this correlative measure, we observed sample values normally distributed around a median of 0 for the 30 gene signature, indicating minimal potential for identifying samples based on their patient-of-origin (Fig. 3a, Supplementary Table 1). Increasing similarity values were observed for all the other signatures relative to the 30 gene signature (Fig. 3a, Supplementary Table 1). In particular, the robustness of the Popovici signature to group samples by patient-of-origin was evident from these analyses. Surprisingly, given the high number of CT and IF patient samples concordantly clustered using the Kennedy signature (88%, Fig. 1) the median value recorded for this signature when challenged with the addition of the LN samples (0.53) appeared to be relatively low compared to that of the Popovici signature (0.73) (Fig. 3a, Supplementary Table 1). In line with the normalized data, the unadjusted similarity matrices for the stem-like (CMS4), Jorissen, Eschrich, Sadanandam, Kennedy and Popovici signatures also confirmed an increased qualitative trend for all signatures compared to the 30 gene stromal-derived signature (Supplementary Fig. 2a–g).

Given the results of the clustering and similarity analyses, we hypothesized that the level of performance observed for each signature would be relative to the proportion of stromal and epithelial genes in each classifier. We also proposed that the Popovici signature genes would be predominantly epithelial tumour cell derived, given the superior performance of this signature in our study. The Popovici signature was developed by examining the transcriptional profile associated with epithelial *BRAF* mutational status using gene expression data from the PETACC[15] stage II/III clinical trial[14]. To test our hypothesis,

we examined median expression values of transcriptional profiles generated from individual tumour cell compartments (epithelial, leukocyte, endothelial and fibroblast)[16] for each signature. In line with our previous study, we observed that the 30 gene signature is predominantly fibroblast in origin (Fig. 3b). Similarly, the stem-like (CMS4), Jorissen, Eschrich and Sadanandam signatures are also dominated by fibroblast-derived genes, providing an explanation for their poor performance due to stromal-derived ITH (Fig. 3b). The Kennedy signature appeared to have a more balanced proportion of epithelial- and stromal-derived (leukocyte, fibroblast and endothelial) transcripts as evidenced from their relative expression values, providing an explanation for its performance in initial clustering analysis (Fig. 3b). Importantly, and in line with our hypothesis, we found that the source of the 64 genes in the Popovici signature was predominantly epithelial in origin (Fig. 3c, Supplementary Fig. 3a).

These results further reinforced the findings of our previous work, in which we reported that cancer-cell extrinsic genes can confound transcriptomics-based patient classification signatures[10], while also suggesting that a classifier based on intrinsic neoplastic gene expression could potentially overcome the confounding factor of infiltrating tumour stroma (Figs 1 and 2). To further test this hypothesis, we utilized the recently published CRC intrinsic signature (CRIS)[17], which was generated by profiling human transcripts from patient-derived xenograft (PDX) tissue. As the original tumour stroma is replaced by mouse stroma in PDX models, stromal-derived gene expression is absent from these human-specific gene expression profiles. Therefore, this approach allows assessment of gene expression originating only from the cancer cells, which could otherwise be masked by extrinsic stromal gene expression[18]. As with the Popovici signature, we confirmed the epithelial nature of the CRIS gene expression signature (Fig. 4a, Supplementary Fig. 3b). Using the DIANA method of clustering expression profiles for the CRIS signature genes, (initially by comparing the CT and IF samples), we demonstrated that 22 out of 24 patient samples (92%) clustered based on patient-of-origin (Fig. 4b,c), the highest concordance of all 8 signatures assessed. Sample clustering of CRIS genes using Euclidean metrics following the inclusion of the additional 24 metastatic LN samples, indicated that the CRIS signature can group samples by patient-of-origin, irrespective of whether the sample is obtained from either primary or metastatic material (Fig. 4d). Interestingly, we identified a 19 gene overlap between the Popovici and CRIS signatures and on examination of these genes, we found that these are predominantly epithelial expressed genes rather than genes expressed in endothelial, leukocyte or fibroblasts (analysis of variance $P < 0.0001$, Tukey's multiple comparison test $P < 0.05$, Supplementary Fig. 2d,e), further reinforcing the intrinsic signature hypothesis.

To directly compare the patient classification results using the published methodologies for both the CRIS and CMS classifiers, we performed sample classification with the random-forest CMS classifier algorithm, alongside the CRIS classifier, which uses a nearest template prediction (NTP) classifier, on our complete cohort. We observed that while CMS classification results in concordant assignment of 9/24 (38%) of patient-matched CT and IF samples, the CRIS classifier concordantly assigns 17/24 (71%) of patient-matched CT and IF samples (Fig. 4e,f). More detailed analysis of concordance between the CT and LN (CMS 29%, CRIS 46%), IF and LN (CMS 21%, CRIS 50%) and the complete multiregional data set ((CT, IF and LN samples)—(CMS 17%, CRIS 42%)) again clearly demonstrated a higher level of agreement using the CRIS classifier in each sub-analysis (Fig. 4e). In addition, given the high level of samples classified as unknown using the CMS random-forest

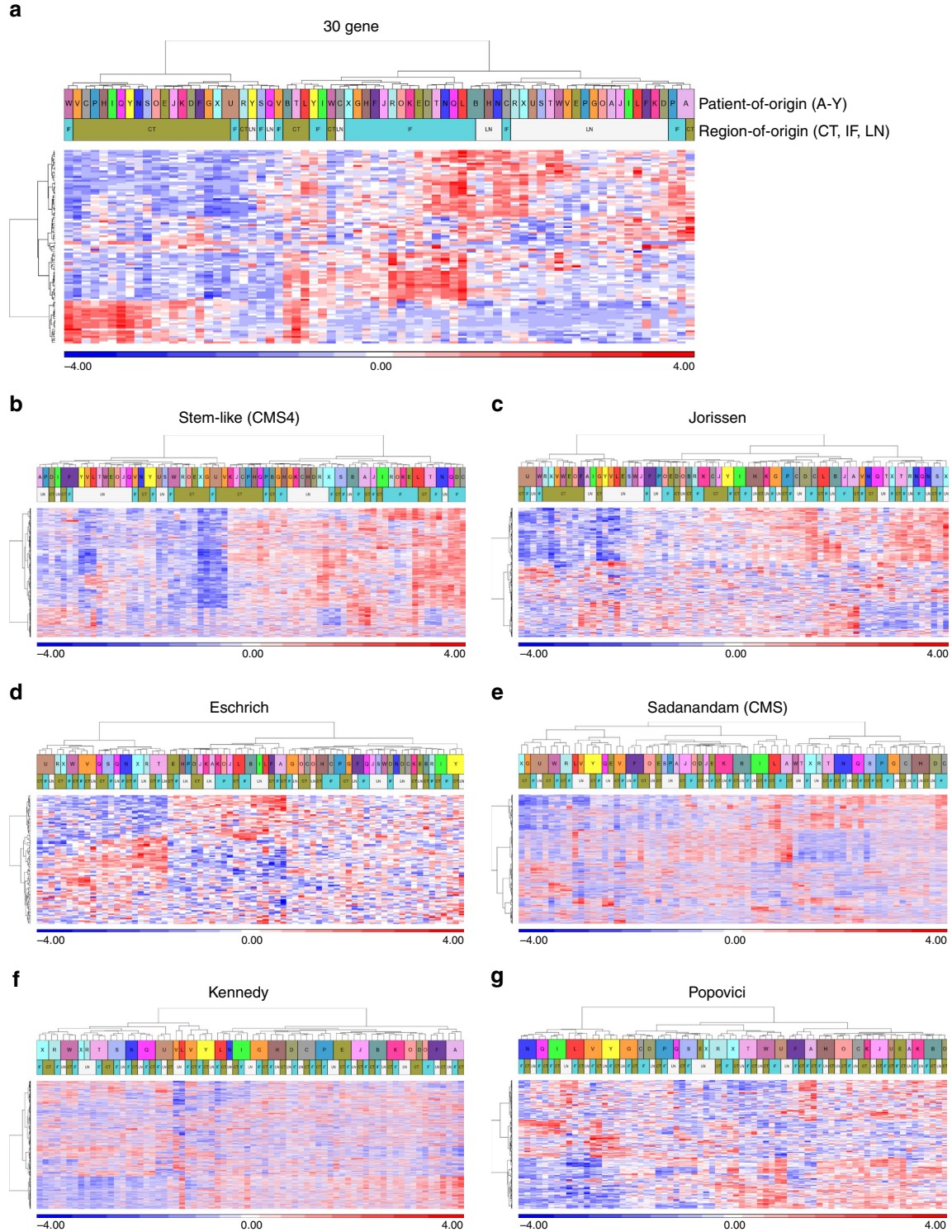

**Figure 2 | Assessment of multi-regional sample clustering using primary and matched metastatic tissue.** (a–g). Hierarchical clustering of our extended patient cohort, including CT, IF and LN tumour tissue, based on semi-supervised expression profiles of genes from the previously published 30 gene signature (**a**) and each individual independent gene signature, namely the stem-like (CMS4) (**b**), Jorissen (**c**), Eschrich (**d**), Sadanandam (CMS) (**e**), Kennedy (**f**) and Popovici (**g**) signatures. Top overlay bar represents colour coded patient-of-origin, labelled A–Y, with the bottom overlay bar representing region-of-origin, CT, green; IF, blue; LN, white.

classifier (UNK), the number of patients with no overlap in subtype classification was 46% for CMS, whereas the value for CRIS was 8% with only two patients displaying no concordant classification in any multiregional samples (Fig. 4e,f). In agreement with the data in Fig. 1a, and in line with our previous work[10], we observed the effect of stromal-derived ITH in our cohort through the differences that we observed in CMS classification, particularly CMS4, of samples according to region-of-origin in the CT, IF and matched LN tissue (Supplementary Fig. 4).

**Combined assessment of patient classification.** Further comparison of the CRIS signature using the patient similarity normalized index as before (Fig. 3a), indicated that the robustness of the CRIS signature (0.62) is ranked higher than all signatures other than Popovici signature using this metric (Fig. 5a, Supplementary Fig. 2h). To further test and compare the data obtained by clustering and similarity analyses, we performed a further semi-supervised clustering approach, whereby we

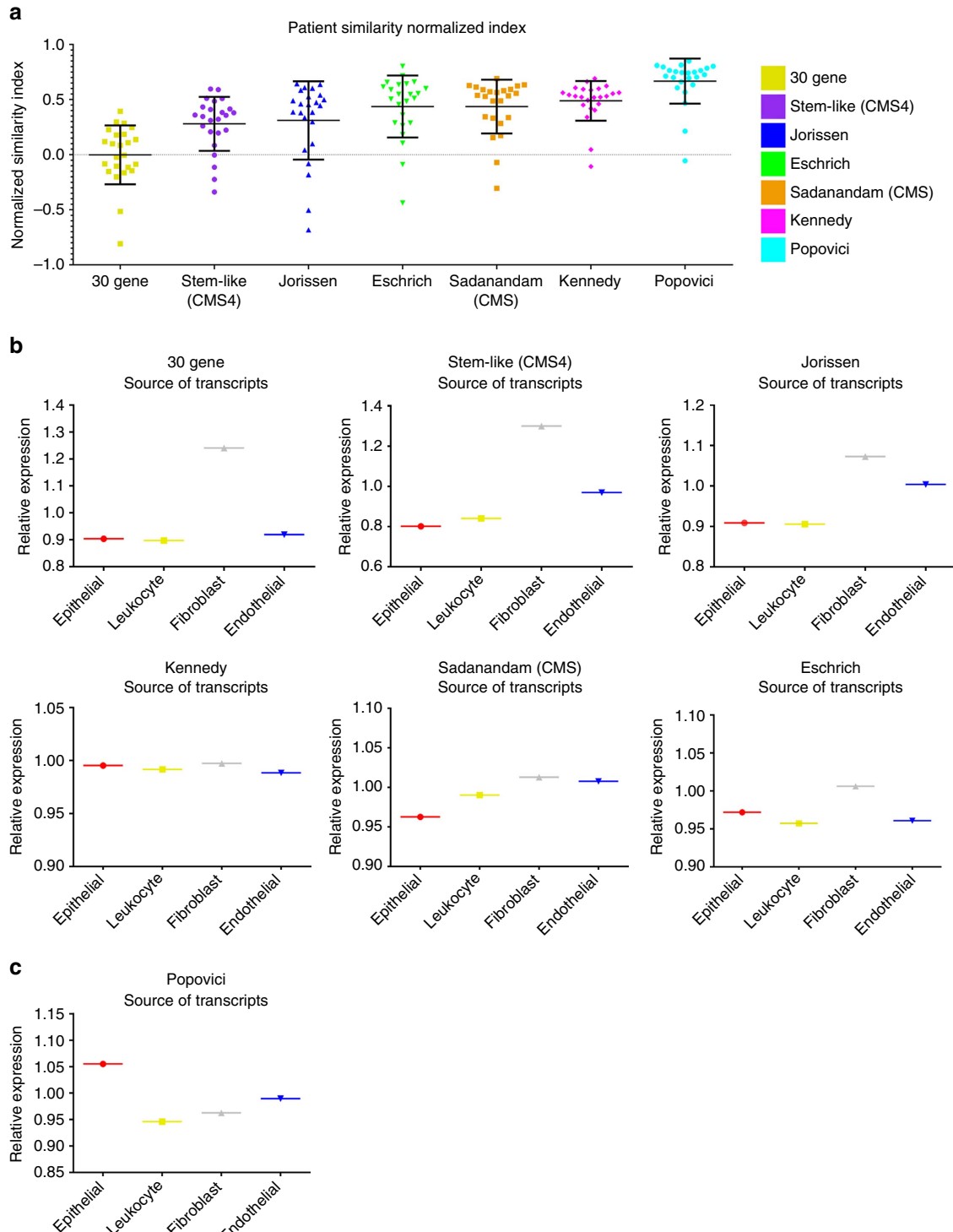

**Figure 3 | A higher proportion of epithelial transcripts enables concordant clustering of patient tumour samples regardless of region-of-origin.** (**a**) Dot plots using normalized Pearson similarity scores for each individual gene expression signature as indicated. Error bars on dot plots represent s.d. values, with median bar. The colour label key for each signature is indicated on the right of this plot. (**b**) Median expression of all probesets annotated to the genes according to the cell-specific source of the transcripts in the 30 gene, stem-like (CMS4), Jorissen, Eschrich, Sadanandam (CMS) and Kennedy signatures using epithelial, fibroblast, endothelial, and leukocyte populations isolated by FACS (GSE39396). (**c**) Median expression of all probesets annotated to the genes according to the cell-specific source of the transcripts in the Popovici signature using epithelial, fibroblast, endothelial, and leukocyte populations isolated by FACS (GSE39396).

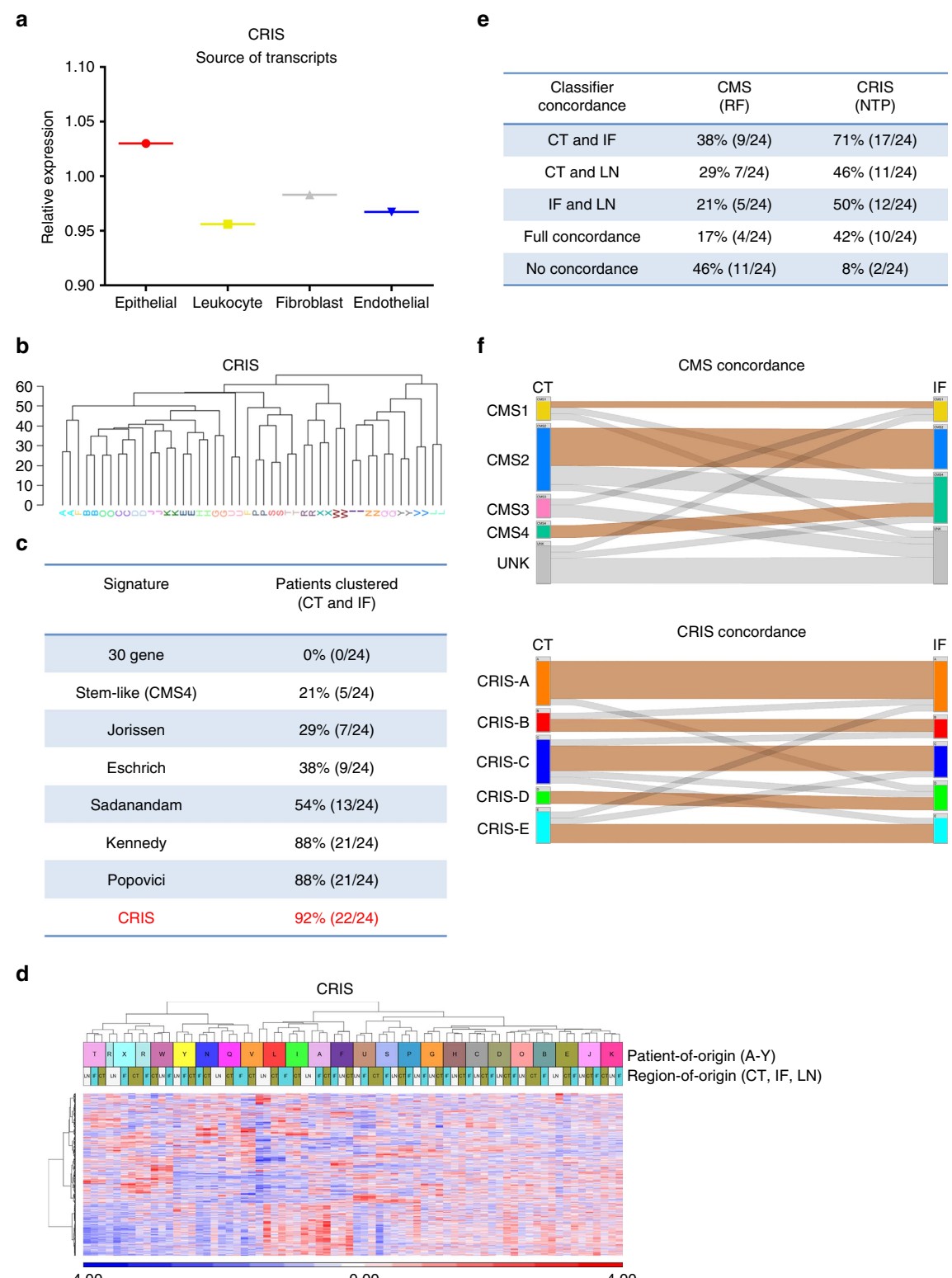

**Figure 4 | The CRC intrinsic signature (CRIS) enables concordant clustering of patient tumour samples regardless of region-of-origin.** (**a**) Median expression of all probesets annotated to the genes according to the cell-specific source of the transcripts in the CRIS signature using epithelial, fibroblast, endothelial, and leukocyte populations isolated by FACS (GSE39396). (**b**) DIANA clustering of CT and IF patient samples based on the gene expression of the CRIS signature. (**c**) Table of concordantly clustered patient samples (as in Fig. 1d) now including the CRIS signature. (**d**) Hierarchical clustering of our extended patient cohort, including CT, IF and LN tumour tissue, based on semi-supervised expression profiles of CRIS signature genes. Top overlay bar indicates patients, bottom overlay bar indicated region-of-origin. (**e**) Table of concordantly clustered patient samples using either the CMS Random-forest (RF) classifier or the CRIS Nearest Template Predictor (NTP) classifier. (**f**) Caleydo (StratomeX) graphical representation of the highest predicted CMS score (CMS1-4, UNK = Unknown assignment) and CRIS subtype (CRIS-A-E) for each sample according to region-of-origin. Concordant subtype assignment of samples is indicated by orange coloured linker, discordant subtype assignment of samples is indicated by grey coloured linker.

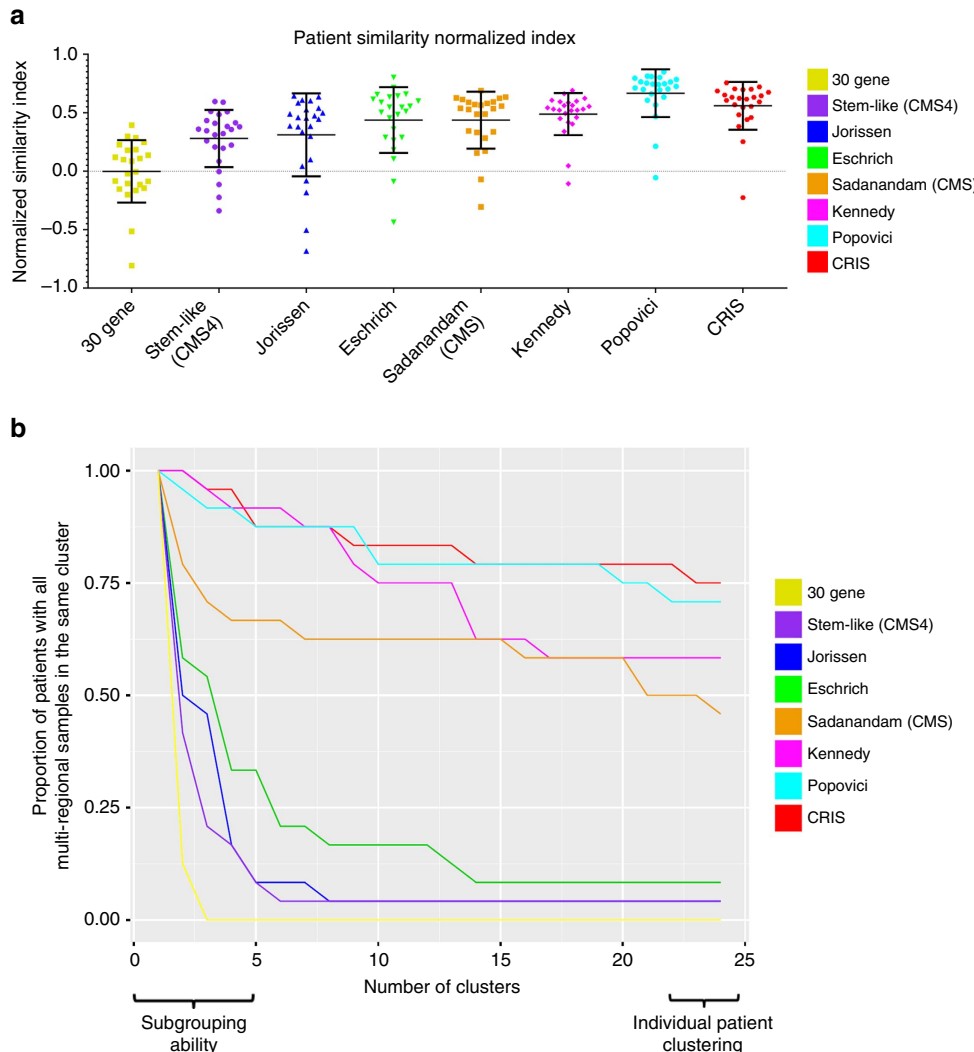

**Figure 5 | Assessment of multi-region sample clustering into concordant subtypes and into individual patient clusters.** (**a**) Dot plots using normalized Pearson similarity scores for each individual gene expression signature (as in Fig. 3a) now including the CRIS signature. Error bars on dot plots represent s.d. values, with median bar. (**b**) Patient group overall ratio plot demonstrating the ability of each individual signature to concordantly cluster patient samples as an indication of confounding transcriptional ITH. For example, the 30 gene signature displays an immediate drop in concordant sample clustering, indicating high levels of confounding transcriptional ITH. The CRIS signature maintains a high level of concordant clustering at both the initial subtype, (left of x axis), and continues to cluster samples according to each individual patient, (right of x axis). The proportion of patients with all samples in the same cluster is measured on the y axis from 1 to 0, relative to the number of clusters on the x axis from 2 to 24. Each continuous signature score is colour coded as outlined in the legend and in the colour label key.

sequentially increased the number of clustering branches, or clades, from 2 (initial subtyping) up to 24 (the total number of patients) for each individual signature. This enabled step-wise measurement of the extent to which patients samples remain correctly clustered together through increasing stratification and demonstrated that the 30 gene, stem-like (CMS4), Jorissen and Eschrich signatures were unable to robustly cluster patient samples into the same subtype, confirming our earlier assessment of the lack of accuracy of these signatures (Fig. 5b). In contrast, these analyses confirmed the ability of the Kennedy, Popovici and CRIS signatures to cluster patient samples into the same initial patient subtype, regardless of the site-of-origin, which is a key requirement for prognostic/predictive signatures (Fig. 5b). The Sadanandam signature[20] displayed an immediate reduction in its ability to maintain multiregional sample clustering according to patient-of-origin, to below that of the Kennedy, Popovici and CRIS signatures. Importantly, while Popovici and CRIS signatures

both maintain their capacity to cluster samples according to patient-of-origin even at the final level of stringency (75% concordance); this analysis further explains the reduced ability of the Kennedy signature for patient clustering using the normalized similarity method (Fig. 5a), as although this signature displays effective subtyping ability in the initial stages, its suitability for the most stringent individual patient clustering ultimately is reduced to almost 50% in our data set, similar to the results of the Sadanandam (CMS) signature (Fig. 5b).

The potential clinical utility of both the Popovici and CRIS signatures for patient classification is further reinforced by their ability to maintain patient-specific clustering throughout this step-wise analysis, with only a minimal decline in their efficiency as the stringency increases (Fig. 5b). By plotting the score recorded for each signature at 2 and again at 24 clusters, the ability of the Kennedy signature for initial assignment into 2 potentially prognostic subgroups is again demonstrated (100% initial subgroup

concordance; Supplementary Fig. 5) although its ability is reduced as the number of patient clusters evaluated, and therefore the stringency, increases. This analysis highlights the robust nature of both the Popovici and CRIS signatures to concordantly cluster samples into both the same initial subgroup and to continue to maintain a high level of concordance in the final patient clusters according to patient-of-origin (Supplementary Fig. 5).

## Discussion

We and others have previously demonstrated how transcriptional-based patient classifiers, such as the CMS[8–10], are affected by tumour sampling region due to changes in the stromal-derived cellular content and region-specific gene expression profiles across the 3D structure of the tumour architecture[10]. The ability of a transcriptional-based signature to consistently classify a patient's subtype even at a metastatic site was posed as one of the challenges which remain to be addressed by Morris and Kopetz recently[18]. Thus, the addition of metastatic tissue to our analysis is highly relevant, as it represents tissue which has undergone the process of EMT, invasion and tumour initiation at the metastatic site. Data presented here further supports our previous work, by confirming that sampling tissue from the invasive regions of a primary tumour increases the likelihood of a tumour being assigned a CMS4 classification. Indeed, in line with the dependence of CMS4 on the presence of fibroblasts/stroma, the ability of either the published random-forest CMS algorithm or the Sadanandam stem-like classifier, (as a surrogate of CMS4), to concordantly cluster patient samples was low across all analyses performed in this study. Importantly, using the published CMS classifier, we observed a high level of unclassified samples in all regions, particularly in the invasive front or metastatic samples, the majority of which were assigned as epithelial-rich CMS2/3 in the centre of the primary tumour, demonstrating the limited clinical utility of CMS classification using tissue from the IF or indeed the metastatic lymph nodes. Thus, the changes in cell type populations in the surrounding TME provide a stringent test of the performance of a published transcriptional signature. Given our experimental design and sample collection, we were able to directly assess individual signatures for their ability to consistently classify patients, regardless of the site from which the tissue was extracted. Our previous data highlighted the extent to which stromal-derived ITH can undermine CMS patient stratification[10]; we now reveal that this confounding issue can also compromise other published transcriptional-based signatures.

Unsupervised clustering of large transcriptomic array data will generally cluster samples according to patient-of-origin. However, classification of patient samples into prognostic/predictive subtypes is generally dependent on small pre-defined gene signatures that can themselves be confounded by region-of-origin of the tissue. When patient stratification is distilled to this level of granularity, subtle changes in the global gene expression profiles at different regions can have profound implication for subtype assignment. Similar to results with our 30 gene signature, the stem-like CMS4 classifier is confounded by region-of-origin when attempting to classify individual patient samples. Signatures such as the Jorissen and Eschrich signatures[11,12] also failed to cluster samples from the same patient into the same tumour subtypes in our dataset. The Sadanandam classifier displayed an intermediate ability to concordantly cluster samples according to patient-of-origin. In summary, these 4 signatures could potentially yield different diagnostic classification results for an individual patient, depending on the site of tissue sampling. In the case of the Kennedy signature[13], we observed highly robust and concordant patient stratification at the initial prognostic subtype stage, fulfilling the purpose for which this signature was designed. However, the confounding effects of transcriptional variation at different regions of a tumour reduce the ability of the Kennedy signature to be applied at an individual patient classifier level.

In contrast, the Popovici signature[14] and the recently proposed CRIS signature[17] precisely cluster samples according to patient-of-origin. We have also demonstrated that both the Popovici and CRIS signatures are enriched for cancer-cell specific intrinsic gene expression. Signatures based on intrinsic genes can thus classify tumours according to the transcriptional variations in neoplastic epithelial cells that may otherwise be difficult to detect among the transcriptional 'noise' generated by stromal cell types within the overall TME, thereby allowing true inter-patient variation to be identified. This method of examining neoplastic signalling in isolation from the surrounding stroma was also proposed by Morris and Kopetz as a way of potentially improving gene expression classification[18]. Our findings confirm the utility of both the Popovici and CRIS signatures for patient stratification and further support the development of intrinsic gene expression signatures as a means to avoid patient misclassification as a result of confounding changes in TME and the stromal, particularly fibroblast, cellular contents in a tumour.

Large individual and collaborative efforts aimed at defining CRC molecular subtypes have confirmed the important role that the stroma plays in patient stratification and prognosis[7,19–23]. These recent findings mirror older traditional pathology-based studies which have shown that high levels of stromal infiltrate provide valuable prognostic information for disease management[24–26]. Identification of a patient's overall tumour subtype, including both intrinsic and extrinsic components within the TME, remains an important tool for both translational research and clinical decision-making. Data presented here do not negate the relevance of these now well-established and clinically informative stromal-based subtypes; rather we have highlighted the potential challenge of robustly identifying a patient's molecular subtype using transcriptional signatures which also capture stromal-derived gene expression. This issue may be particularly problematic when patient stratification decisions are based on the often small amounts of primary or metastatic biopsy tissue which are available for analysis in prospective clinical trials, where control over region-of-origin and stromal content of the tissue samples is limited. Data presented here indicate how gene expression signatures which are predominantly derived from neoplastic epithelial cells can alleviate such confounding issues, enabling more robust patient classification regardless of the region(s) from which the tissue has been extracted. These findings may facilitate better transcriptional-based tracking of primary and metastatic disease from an individual patient and may ultimately help in the development of better genomic tools for stratification around patient prognosis or indeed prediction of outcome from therapy. This level of disease tracking and biological understanding is particularly important for the increasing numbers of patients diagnosed with early stage disease. Dukes A/B accounts for up to 71% of bowel screen-detected CRC cases[27], where prevention or informed treatment following disease progression can make a significant impact to cancer survival rates.

The platforms used in the generation of the gene signatures in this study include Affymetrix and custom cDNA arrays, alongside next generation sequencing (NGS) technology. Inevitably, when comparing the utility of these signatures, there will be some cases when individual genes/probes are not universally represented across all platforms, resulting in gene dropout. To ensure that this dropout was minimized, we utilized consensus 'core genes' for the signatures (detailed in Supplementary Data 1) and as defined previously by Sanz-Pamplona et al.[4] to enable cross-platform

validation of signatures. Comprehensive updating was performed on the DSA array annotation to include recently annotated transcripts and additional gene aliases from Entrez ID, Unigene ID, HUGO and RefSeq was also performed. Despite these efforts, inevitably no match was found for a small number of genes in a number of the signatures but the levels of gene drop-out were minimal. As such, the genes used in our study are as a representative of the original signatures. Importantly however, the use of core genes has previously been shown to maintain the prognostic value of the original signatures studied.[4]

The approach we have taken here is not aimed at validating the outcome predicted for each signature; rather we aim to stringently test the ability of each signature to concordantly cluster and, in turn, identify samples from different regions-of-origin from the same patient, without modification or weighting of the signature genes. Despite the overall robust nature of the Popovici and CRIS signatures observed in this study, our cell-lineage specific analysis indicates that there are still some residual stromal-derived transcripts in the Popovici and CRIS signatures, which may explain the small drop in patient-specific clustering at the most extreme statistical limits of our analyses. In addition, although the ability of a signature to robustly cluster patient samples regardless of region-of-origin is important for a clinical test, this characteristic on its own is not sufficient for it to be a clinical useful tool. Clinical utility is only possible if the cancer-cell intrinsic signatures themselves also provide additional prognostic/predictive or important clinical information beyond currently utilized methods. The generation of patient classifiers and the validation of their clinical utility in large retrospective cohorts should attempt to address the confounding variable of changes in tumour content as a first step in refining signatures with prognostic or predictive potential. The generation of a large data repository of clinical samples containing profiles from multiple regions of each tumour, similar to the datasets that we and others have assembled[10,28], may provide an important resource to help improve the identification of robust clinically relevant transcriptional signatures. Such a repository would also allow a more in-depth analysis of ITH, not only within a patient, but also at each site in the primary or metastatic lesion using increased numbers of samples at each tumour site.

As in our previous study[10], we show here that the confounding issue of stromal derived intratumoural heterogeneity is also evident in a number of clinically-relevant transcriptional signatures. However, we now extend this observation to demonstrate that gene signatures based on epithelial cancer-cell intrinsic gene expression result in significantly more robust and reliable stratification of CRC patients, when compared to stromal-dependent signatures such as CMS. The refinement of patient classifiers through a well-informed pipeline, including stromal-derived ITH as an inherent covariate as demonstrated in this study, has the potential to translate this biology-driven research into informative clinical signatures that can be reliably employed as diagnostic, prognostic or predictive tools.

In conclusion, our data support the clinical evaluation of signatures based on intrinsic gene expression, such as the Popovici and CRIS signatures, which are largely unaffected by the confounding variable of stromal-derived ITH. These cancer-cell intrinsic signatures have the potential to be used clinically to inform precision medicine decisions, ultimately leading to improved patient outcomes.

## Methods

**The initial dataset.** The patient samples employed in this study were as described previously[10]. Briefly, we selected fresh primary tumour colon resection specimens from 25 patients where we had sufficient high quality material to extract RNA from the regions of interest. Each patient was labelled alphabetically from A to Y.

This resulted in 75 transcription profiles, 25 patients' samples at 3 regions per sample, namely the Invasive Front (IF), Central Tumour (CT) and Lymph Node (LN). These 75 samples were analysed on the CRC disease specific array (DSA) platform (Almac Diagnostics Ltd). This CRC DSA has a total of 61,528 probe sets and encodes 52,306 transcripts, which we have determined to represent 15,273 annotated genes. The CRC DSA provides robust profiling of RNA derived from FFPE tissue samples compared to Affymetrix microarrays, due to its disease specific content (~20,000 transcripts are unique to the CRC DSA) and enhanced ability to detect degraded RNA by using a 3′-based probe design.

Following the generation of transcriptional data, we used distribution analysis, hierarchical clustering (condition tree) and principal components analysis (PCA) to assess the quality of the data before analysis. The distribution of sample data (histogram of normalized intensity values) should approximate a normal distribution and should be free of outliers, demonstrated through hierarchical clustering and decomposition techniques (PCA and/or MDS). Hierarchical clustering (condition tree) shows the relationships among the expression levels of samples, allowing identification of any spurious samples. From the initial QC analyses of the microarray data, we identified 2 samples as outliers from the PCA and clustering tree methods (Supplementary Fig. 6) which were then removed from the analysis. Subsequent alignment of patient ID codes revealed that these 2 samples originated from the same patient; therefore the remaining sample from that patient (Patient M) was also removed from further downstream analyses. Our approach ensured that each patient was represented by samples from 3 different regions of the tumour (which we refer to as multi-region samples). The final processed cohort therefore contained 72 transcriptional profiles; 24 IF, 24 CT and 24 LN tumour data deposits.

**Classification of CMS and CRIS molecular subtypes.** The consensus molecular subtype (CMS) for each region profiled was determined using the random forest classifier described by Guinney et al.[7] The CMSclassifier R package was downloaded from github (https://github.com/Sage-Bionetworks/crcsc) and scripts executed in R studio version 3.3.1. The colorectal intrinsic subtypes (CRIS) were determined using the nearest template prediction (NTP) classifier, available from GenePattern (https://genepattern.broadinstitute.org/gp/pages/login.jsf). The thresholds used for statistical significance were generated using Benjamini-Hochberg–corrected false discovery rate (BH.FDR) < 0.2 (ref. 29).

**Gene expression signatures employed in the study.** To avoid variations inherent during probeset-to-gene annotation conversion from signatures developed using different platforms, we utilized previously published consensus gene lists for each signature to be evaluated, as implemented by Sanz-Pamplona and colleagues[4]. We then cross-referenced this consensus gene list with the genes from our CRC DSA Affymetrix microarray platform, resulting in a list of signature genes used in this study (Supplementary Data 1). To further minimise any gene dropout from the signatures used in this study, we updated the annotation for the CRC DSA to current HUGO nomenclature by matching the Entrez Gene ID of each probe to HUGO gene symbols obtained from the HUGO Gene Nomenclature Committee (HGNC) dataset file: 'hgnc_complete_set' as available on 31st January 2017 from the EBI ftp server: (ftp://ftp.ebi.ac.uk/pub/databases/genenames/new/tsv/hgnc_complete_set.txt)

In addition, we have used a combination of Entrez ID, Unigene, HUGO and Refseq to comprehensively update the annotation file for the DSA array. Probes without Entrez Gene IDs that had a RefSeq or Unigene ID in the original annotation were assigned an Entrez Gene ID using the NCBI Batch Entrez tool. Probes that hadn't any identifiers apart from a gene symbol were updated by matching that gene symbol to a previous gene symbol or alias found in the hgnc_complete_set file. Despite these efforts, inevitably no match was found for a small number of genes in some signatures and they were lost for subsequent analysis. The overall levels of gene dropout were minimal; the signature with the highest dropout (Eschrich; 10.3% dropout) was due to a loss of only three genes missing from the core gene signature. Given this limitation, the signatures used in our study can only serve as a representation of the original signatures. Importantly however, our study is focussed on dissecting the cellular source of the core genes related to their ability to robustly cluster patient samples to outline parameters that could potentially improve future signature/classifier design. Each individual signature, in its original form, has demonstrated prognostic value and was selected as they represent the most clinically relevant and widely employed signatures to support the aims and findings in this study.

The 30 gene signature was previously generated[10] as a contrast between CT and IF in this cohort (Supplementary Data 1).

The Eschrich et al.[12] signature was developed using 78 pre-selected frozen colon tumours samples based on their OS survival status at 36 months, good > 36 months survival (n = 30), poor < 36 months survival (n = 48) across Dukes stage B, C and D. Three adenoma samples were included within the good prognostic group. Transcription profiles were generated on cDNA arrays. Using a leave-one-out cross validation approach, a 43 gene signature was developed (Supplementary Data 2).

The Jorissen et al.[11] signature was developed from transcriptional profiles from 553 colorectal samples using Affymetrix HG-U133Plus2.0 GeneChip arrays. This cohort consisted of 293 fresh frozen tumour specimens, with the remaining

260 samples being identified from publically available gene expression data. Differential gene expression (DEG) changes were identified between Dukes stage A and D in both the in-house and public datasets. Additional analysis was also undertaken between primary stage D and metastatic tissue, to develop 'metastasis associated genes' which through 2-fold cross validation were used to generate a 163 gene classifier which could stratify the stage B and C samples into prognostic subtypes. The prognosis signature was enriched for downregulated immune response genes and upregulated cell signalling and extracellular matrix genes (Supplementary Data 3).

The Sadanandam et al.[20] signature was developed from the transcriptional profiles from 445 primary CRC resections (GSE13294 and GSE14333) using Affymetrix HG-U133Plus2.0 GeneChip arrays. Consensus-based non-negative matrix factorization was used to define five molecular subtypes. The 786 subtype-specific signature genes were then identified using significance analysis of microarrays (SAM) and prediction analysis for microarrays (PAM). This signature was used as a surrogate for the CMS classification. The 207 genes associated with classification of the stem-like subtype from the original Sadanandam et al.[20] signature were used as a surrogate for CMS4 (Supplementary Data 4).

The Kennedy et al.[13] signature was developed from a cohort of stage II FFPE colon cancer tumours, preselected for samples based on: risk of recurrence within 5 years of surgery; disease recurrence ($n = 73$) or no recurrence ($n = 143$). Transcriptional data was generated on the Almac Colorectal Cancer DSA platform. Using cross validation, a 634 probeset signature was developed which identified a prognostic subtype in stage II, which was further validated in an independent cohort of 144 samples. This signature has subsequently been validated in an additional stage II clinical trial cohort[30] (Supplementary Data 5).

The Popovici et al.[14] signature was developed using 668 stage II/III FFPE colon cancer tissue samples from the PETACC-3 phase III clinical trial investigating adjuvant treatment and disease-free survival in colon cancer[15]. Transcriptional profiling was performed on the Almac Colorectal Cancer DSA platform. A 64 gene classifier was developed using multiple top scoring pair's method and cross-validation based mutational status which identifies samples with signalling similar to BRAF mutant tumours. This signature was found to identify a poor-prognostic subtype, based on overall survival in the training dataset and independent validation cohorts (Supplementary Data 6).

The colorectal intrinsic signature (CRIS)[17] was developed using 515 patient-derived xenograph tumours from liver metastasis tissue extracted from 244 patients. Implantation of patient tissue into murine models resulted in the replacement of non-transformed stromal cells with murine stroma. The neoplastic epithelial component of the original tumour tissue is preserved. Transcriptional profiles were generated using Illumina human-specific 48k gene chips. The resulting transcripts were tested for cross-species contamination to ensure that they originated from cancer cell intrinsic expression only. Following non-negative matrix factorization and cross-validation, a final 565 gene classifier was developed which could stratify a number of in-house and independent datasets into subtypes with prognostic utility. The CRIS signature has been shown to have improved prognostic significance compared to the CMS classifier in a number of independent datasets, by clustering patients based on the subtype of their intrinsic epithelial expression profile, rather than the tumour bulk tissue which includes the non-neoplastic TME (Supplementary Data 7).

For all analyses, we have used the gene sets as detailed in Supplementary Data 1–7, followed by hierarchical clustering or Pearson similarity assessment with standard methodology. The gene sets did not undergo any adjustment or weighting and were not modified with any algorithm.

**DIANA clustering methodology.** As outlined above, patients were assigned an alphabetical label A–Y; patient sample M was removed as an outlier before analysis. Divisive analysis clustering (DIANA) was performed on the expression values of the patient matched IF and CT samples, ascertaining to each of the gene signatures on default settings. This was completed using the 'cluster' package in R statistical environment (v3.2.3). The DIANA method is particularly suited to test inter-patient heterogeneity in sample pairs, as it continuously splits samples into two clusters until it reaches single samples, which allows an assessment of both the final patient clustering and the extent to which an individual signature could be undermined by stromal derived transcriptional ITH. Resulting dendrograms were assessed for patient clustering and values were tabulated.

**Transcriptional clustering analysis.** Partek Genomics Suite software, version 6.6; 2016 Partek, Inc., St Louis, MO, USA., was used for independent dataset analysis. For the purpose of clustering, data matrices were standardized to the median value of probe sets expression. Standardization of the data allows for comparison of expression levels for different probe sets. Following standardization, two-dimensional hierarchical clustering was performed (samples × probe sets/genes). Euclidean distance was used to calculate the distance matrix, a multidimensional matrix representing the distance from each data point (probe set-sample pair) to all the other data points. Ward's linkage method was subsequently applied to join samples and genes together, with the minimum variance, to find compact clusters based on the calculated distance matrix.

**Normalized pearson similarity scoring.** The Pearson correlation coefficient was used to define the ratio between the covariance and the standard deviation of multi-region samples for each individual patient. By generating a score for each sample compared to each other sample, this method allowed us to build a matrix based on an enumeration of the similarities of all three samples (IF, CT and LN) for each individual patient. Increased scores indicate that samples display a higher similarity with other matched region-specific samples from the same patient. As the standard Pearson method allows direct correlation of one sample to another, we wished to test if each individual patient score was higher than that observed across all of the samples. To this end, we used a normalized method, which calculates the relative similarity between the three samples from the same patient, compared to their similarity to samples from all other patients within our dataset, from a score of 0 (no increased correlation of patient matched samples compared to samples from different patients) to 1 (maximum correlation of patient matched samples compared to all other samples).

**Patient group overall ratio.** We used sequential analyses to give an assessment of the ability of each signature to both cluster patient samples into the same higher-order prognostic/predictive subtype, followed by the ability of the signature to robustly differentiate and cluster primary and metastatic samples according to patient of origin. The various signatures indicated above were used to cluster the data, using hclust, with Ward's linkage and Euclidean distance metric. The resultant dendrogram was then analysed using the cutree function to extract the group membership, as the number of groups is sequentially increased, from 1 to 24 (the number of patients). At each level, the Patient Group Overall Ratio (PGOR) was calculated as: PGOR = Number of Patients Grouped in the Same Cluster/Total Number of Patients, that is, the PGOR = 1 if all samples for all patients are found in the same cluster at a particular level, and PGOR = 0 if none of the samples group together consistently. The evolution of the PGOR was plotted against the number of clusters, showing the consistency of patient clustering.

**Data availability.** Our transcriptional data and updated annotation files, alongside patient and region identifiers has been uploaded to the NCBI Gene Expression Omnibus (GEO) repository (http://www.ncbi.nlm.nih.gov/geo/) and is available under accession numbers GSE95109 and GPL23083. For testing of the cell lineage-specific source of the transcripts, gene expression profiles from an independent CRC dataset were downloaded from NCBI Gene Expression Omnibus (GEO) (http://www.ncbi.nlm.nih.gov/geo/) under accession number GSE39396. The GSE39396 dataset contains microarray profiles from fresh colorectal specimens where FACS has been used to divide cells into specific endothelial (CD45$^+$EPCAM$^-$CD31$^-$FAP$^-$), epithelial (CD45$^-$ EPCAM$^+$CD31$^-$FAP$^-$), leukocyte (CD45$^-$EPCAM$^-$CD31$^+$FAP$^-$) and fibroblast (CD45$^-$EPCAM$^-$CD31$^-$FAP$^+$) populations before microarray profiling. Plots based on transcriptional data were plotted using GraphPad Prism version 5.03 for Windows, GraphPad Software, La Jolla CA, USA, www.graphpad.com. In addition, for the comparison of CRC sample classification by the Sadanandam signature[20] and the CMS, gene expression profiles under the accession number GSE14333 were downloaded from NCBI GEO. This data set contains the transcriptional profiles of 290 primary colorectal cancers using Affymetrix HG-U133Plus2.0 GeneChip arrays. Patient samples defined as 'unknown' by CMS classification in the original Sadanandam study cohort were removed from our CMS analysis. The figure based on this data was created using Caleydo (StratomeX) version 3.1.5 for Windows. All data utilized in this manuscript are available from the corresponding author on request.

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

## Acknowledgements

This work was supported by The Entwistle Family Travel Scholarship (PDD), CRUK/MRC Stratified Medicine Programme (S:CORT; P.D.D., M.A., S.R., T.M., P.G.J. and M.L.) and 2 separate CRUK PhD studentships (A.R. and D.G.M.) and (A.M.B.M. and M.L.).

## Author contributions

P.D.D. conceptualized the research, performed experiments, analysed and interpreted data and wrote, revised and finalised the manuscript. M.A., P.G.O., A.C.R. and A.M.B.M. performed experiments, analysed data and edited final figures. S.R., T.M., P.G.J., S.S.M., D.B.L. and E.K. provided expertize and intellectual input to the overall project. E.K. provided patient samples and transcriptional data. D.G.M. and M.L. conceptualized the research, interpreted data, edited and finalised the manuscript. All authors read and approved the final manuscript before submission.

## Additional information

**Competing interests:** P.G.J.: Previous Founder and Shareholder of Almac Diagnostics; CV6 Therapeutics: Expert Advisor and Shareholder; Chugai Pharmaceuticals: Consultant. The remaining authors declare no competing financial interests.

