## [Peer Review File · Nature Communications]

Reviewers' Comments:

Reviewer #1 (Remarks to the Author)

A large body of work has emerged based on tumor expression profiling for prognostic and predictive purposes. This has recently culminated in a consensus molecular subtyping for Colorectal Cancer CRC. Dunne et al wonder to what extent heterogeneity of a tumor and a different sites incl the metastatic site, impact this subtyping. 2 sites of the primary tumor (central and Invasive front) and a lymph node mets is used in the analysis of 24 patients (= #72 samples). Signatures of different publications are used.

The authors conclude that some signatures are more subtype prone and other more biopsy site prone.

This re-analysis of existing data is certainly of interest to a broader community.

The paper is well written, although not equally clear what has been done exactly (see below). Algorithm and public data accessibility would be required.

I regret that the CMS subtypes are not included much in the paper. Is there a reason why this was not depicted in the Figures (ie Figure 2, Suppl fig2 and fig 4). this would have given insight in the impact of various biopsies on the CMS.

In a previous paper in Clin Cancer Res. 2016 Aug 15;22(16):4095-104. doi: 10.1158/1078-0432 the authors already present and analyse much of the data presented including invasive front, central tumor, and lymph node regions. It is unclear to me what the increment in new data in this paper really is.

The added value of CRIS clustering is not clear to me, beyond the other clustering approaches

It is not entirely clear to me if the authors indeed use the gene set of the other signatures/papers, or if they use a combination of gene sets and algorithms for the results in for example Figure 2.

The title of the paper suggest it would handle intra tumoral heterogeneity (ITH) however is limited to a 1 central and 1 invasive front biopsy (two biopsies of one tumor). Otherwise a (clonal) tumor at a different site (lymph node) is assayed. Additional biopsies, ie additional primary tumor sites or an additional (invasive front) biopsy of metastatic tumor could have been included or discussed.

I have not found the original (raw) array expression data publicly available, only some differential data in Clin Cancer Res. 2016 Aug 15;22(16):4095-104; This despite the fact that the authors do make use of complete data publicly deposited by other groups in the GEO database.

Text in many figures is too small, in particular the CMS-subtypes in Figure 2A, and any of the presented clusters.

Colors (like the grey and beige) or the CMS subtypes are not described in the legend of Figure 2, which makes the figure unclear.

Minor: the discussion starts with a repetition of what was said in the intro (line 221). What actually happens in the QC (that makes one of the data sets drop out) is unclear (line 315)

Reviewer #2 (Remarks to the Author)

The manuscript by Dunne et al. addresses the challenge caused by intra patient heterogeneity in

colorectal cancer to reliable classification by gene expression signatures. The authors have analysed RNA from 24 patients where they have measured gene expression by a CRC specific array from three samples per patient. That is, samples from centre of the primary tumour, its invasive front, and from a lymph node metastasis. Interestingly, and perhaps not surprisingly, signatures that have dealt with patient intrinsic expression produced more stable results across samples from the same patient. The extent of clustering based on patient-of-origin vs. region-of-origin was assessed for seven signatures, four clinically relevant (prognostic/predictive) signatures (Jorrenson, Eschrich, Kennedy, Popovici), two biologically subclassification signatures (CRIS, CMS) and a seventh signature, developed by the authors (30gene), as a positive control to be informative for the region-of-origin.

The results and conclusions of the papers are novel, interesting and important. Although I am not a statistician as such, I also find the types of analyses to be relevant and well documented. Since there are gene expression classifiers being proposed for several different types of cancer, such an evaluation of classifiers is of interest also to a wider (non-colorectal cancer researcher) audience. I liked reading this paper, and the text description of the results was convincing. However, I also found some major issues in the presentation of the results, which need to be addressed.

Major remarks:

The manuscript include many relevant analyses and plotted results from several gene expression profiles, but consistently shows results for only one or a few of these in various figures and tables. I understand that there may be space restrictions in the journal, but it would be a good reference to include similar analyses and plots for all seven signatures - at least as supplementary figures. Some (but not all) examples are the following:

It is confusing that the CMS1 to 4 classifications are not included in many of the tabulations and figures where the four so-called clinically relevant signatures are evaluated and compared. E.g. why is the CMS1-4 and CRIS not included in e.g. Figures 1C+D and CMS in Figures 3 and 4, Supplementary Figures 2 and 3, Supplementary Table 2, and in the Methods description? I am of the opinion that this should be evident from the manuscript text, and not only in a rebuttal letter.

Along the same line: CMS4 is scrutinised in Fig 1A, but why are similar plots not made for CMS1-3 and all the other signatures? At least such plots should be available as a Supplementary Figure. Without these plots as comparison/references, the Figure 1A seems useless as a proof of the intra-patient variation of CMS4 samples.

Further, the Caleydo plot in Fig 2A is great to visualize the CMS intra-patient heterogeneity, but it would be much more informative by showing such a plot also for CRIS, which supposedly have better intra-patient concordance. And perhaps also include similar plots for the other signatures (in supplementary figs)?

In addition to the seven signatures evaluated here, there are also several published signatures which are not dealt with. The authors should better describe why they chose the particular signatures, and it would be interesting to know why other signatures were not chosen. Review articles have been published that go through most published/proposed signatures, which can serve as basis for the authors to list all signatures and why they did/didn't include them in the paper.

The authors write that the CRC disease specific array they used is produced by Almac. But more specifics should be included in the paper of what the content of this array is (e.g. number of probes and genes) in total and in relation to each of the described signatures. That is, the number and proportion of genes included for each of the signatures - as well as some of the principles behind how the genes in the Almac were selected.

The gene expression data was not available for me to review. I think this is fine, but the authors should describe how they will deposit the newly generated expression data (72 samples). The site

(public repository) and accession number should be included in the manuscript.

Minor comments:

P15. It is described that there were originally 25 patients, A to Y. Then that patient M was removed, so there were 24 patients left for further analyses. Then the name range is changed to A to X in p.5 line 100, although it is used A to Y in figures, leaving M out.

In Supplementary Table 1, cell E87, a gene symbol has been formatted as a date ("mar.03")

Reviewer #3 (Remarks to the Author)

In this study, Dunne et al build on their prior investigation of the impact of intra-tumor heterogeneity (ITH) in CRC on molecular subtyping (Clin Can Res 2016). Given the significant genomic, phenotypic and cellular ITH in diverse solid tumors, including CRC, this is an important issue. Indeed, previous studies have demonstrated that different regions of the same tumor can be classified into distinct molecular subtypes, posing challenges for precision medicine. In addition to genetic heterogeneity, immune cell infiltration and stromal cells contribute to apparent ITH resulting in a reduction in the purity of the tumor cell population and, signal attenuation in various molecular assays. However, it is increasingly appreciated that stromal/immune cell infiltration is an inherent feature of the tumor microenvironment and enriched in specific molecular subgroups.

Here, the authors extend their previous analysis of multi-region expression data for 24 primary CRCs (central tumor and invasive front) and matched lymph node mets on a custom CRC array. They report that the gene expression data cluster by patient rather than tissue of origin. They further note that cancer-cell intrinsic signatures tended to result in more robust patient-specific classifications than other signatures, thereby overcoming the issue of ITH.

While the manuscript addresses an interesting topic, it is not clear that this represents a significant advance over the authors previous work on this topic. I also have concerns regarding aspects of the methods, and therefore the claims that are asserted.

Specific comments:

- Much of the manuscript is focused on the evaluation sample clustering using different signatures. However, given the extensive variation in gene expression across patients, it does not seem surprising that samples from the same patient tend to cluster more tightly than samples from different patients of similar histology/morphology even in the face of differences in stromal composition. Moreover, most gene expression arrays include probes with SNPs that can result in SNP-expression associations. While it is not clear as to how many such probes exist on the custom array employed here (details on tis are scant), this could also contribute to within patient signals.

-The authors also evaluate the classification of LN mets relative to the primary. While mets may harbor a different stromal signature, they are nonetheless expected harbor exhibit similar expression profiles to the primary (as has been shown in breast cancer). Hence, the finding that subtypes are generally similar between primary and LN met is not particularly surprising.

-In terms of signature development, there is no discussion as to whether all genes included in the original signatures are represented on the custom array employed here, . If there was gene dropout (as seems to be the case based on Table S1), this must be noted and the impact of this information loss on the performance of the published signature evaluated (for example with existing public data).

-Several CRC signatures are evaluated in this manuscript including a CRC intrinsic signatures (CRIS). However, there is no further comparison with the consensus molecular subtypes (as was

done in the Dunne 2016 paper). This should be included as additional comparisons are presumably made here (for example comparison of primary tumor regions and matched lymph node metastases). The authors further assert that the Popovici and CRIS signatures have clinical utility since they retain patient-specific clustering. However, this does not equate to clinical utility. Rather, this would necessitate that the signatures provide additional prognostic value beyond established clinical covariates. This is not assessed in this manuscript, and the claim must therefore be toned down.

-The text emphasizes that CRIS enables concordant patient sample clustering. However, this is also true for the Popovici signature; therefore the discussion of the relative merits should be more balanced.

Reviewers' comments:

Reviewer #1 (Remarks to the Author):

A large body of work has emerged based on tumor expression profiling for prognostic and predictive purposes. This has recently culminated in a consensus molecular subtyping for Colorectal Cancer CRC. Dunne et al wonder to what extent heterogeneity of a tumor and a different sites incl the metastatic site, impact this subtyping. 2 sites of the primary tumor (central and Invasive front) and a lymph node mets is used in the analysis of 24 patients (= #72 samples). Signatures of different publications are used. The authors conclude that some signatures are more subtype prone and other more biopsy prone.

This re-analysis of existing data is certainly of interest to a broader community. The paper is well written, although not equally clear what has been done exactly (see below). Algorithm and public data accessibility would be required.

- 1) I regret that the CMS subtypes are not included much in the paper. Is there a reason why this was not depicted in the Figures (ie Figure 2, Suppl fig2 and fig 4). this would have given insight in the impact of various biopsies on the CMS.*

We fully agree with Reviewer #1 that a comparison with CMS was necessary to give further insight into the impact of multi-regional sampling in this study. The CMS subtypes were initially not included in each analysis as, in contrast to other gene classifiers and signatures used in this study, the methodology of CMS classification used in the original study (Guinney et al) is with a random-forest algorithm, making it unsuitable for a direct signature comparison. In order to include a suitable CMS assessment in every analysis throughout our study, we firstly provide a clear demonstration of the utility of the *Sadanandam* CRCA gene signature (Sadanandam et al 2012 Nat Med) as a surrogate marker for CMS (Supplementary Figure 1). In

addition, we also include the specific Stem-like classifier as a surrogate for CMS4 specifically, which we have previously proposed to be the classification subtype that is most prone to variation due to fibroblast content (Dunne CCR 2016). Following the demonstration of the utility of the surrogate signatures, we now include a comprehensive assessment of both CMS and the more specific CMS4 (Stem-like) for each subsequent analysis (Figure 1, 2, 3, 4, 5 and Supplementary Figure 1, 3, 4 and Supplementary Table 1 and 2).

- 2) *In a previous paper in Clin Cancer Res. 2016 Aug 15;22(16):4095-104. doi: 10.1158/1078-0432 the authors already present and analyse much of the data presented including invasive front, central tumor, and lymph node regions. It is unclear to me what the increment in new data in this paper really is.*

In our previous study, we highlighted a potentially confounding problem associated with the CMS classifier, in particular with CMS4, which is heavily reliant on stromal-derived genes for patient classification. Our analysis showed that using CMS, patients could simultaneously be classified into multiple different prognostic subtypes dependent on the “region-of-origin” or their biopsy/tissue sample; in doing so our study identified an important issue that must be considered.

In this current study, we show that the confounding issue of stromal derived intratumoural heterogeneity is also evident in a number of clinically-relevant signatures, but most importantly we now demonstrate that gene signatures based on epithelial cancer-cell intrinsic gene expression result in significantly more robust and reliable stratification of CRC patients compared to stromal-dependent signatures such as CMS. This is a significant advance on what we demonstrated in the Clinical Cancer Research paper.

- 3) *The added value of CRIS clustering is not clear to me, beyond the other clustering approaches.*

The CRIS classifier was recently demonstrated to have improved prognostic value compared to the CMS classifier and currently utilised clinical factors. This data was presented at the AACR annual conference in 2016 and the abstract is referenced (Bertotti et al 2016). As our initial results indicated that cancer cell intrinsic (epithelial) gene expression signatures may be less prone to variation due to stromal intratumoural heterogeneity, we tested this hypothesis (Figure 4, 5 and Supplementary Figure 2, 3, 4,).

- 4) *It is not entirely clear to me if the authors indeed use the gene set of the other signatures/papers, or if they use a combination of gene sets and algorithms for the results in for example Figure 2.*

We have used the gene sets (detailed in Supplementary Table 1), followed by hierarchical clustering or Pearson similarity assessment with standard methodology. The gene sets did not undergo any adjustment or weighting and were not modified with any algorithm. This has been further detailed in the Results and Methods sections.

- 5) *The title of the paper suggest it would handle intra tumoral heterogeneity (ITH) however is limited to a 1 central and 1 invasive front biopsy (two biopsies of one tumor). Otherwise a (clonal) tumor at a different site (lymph node) is assayed. Additional biopsies, ie additional primary tumor sites or an additional (invasive front) biopsy of metastatic tumor could have been included or discussed.*

We have now included a section on this topic in the Discussion.

- 6) *I have not found the original (raw) array expression data publicly available, only some differential data in Clin Cancer Res. 2016 Aug 15;22(16):4095-104; This despite the fact that the authors do make use of complete data publicly deposited by other groups in the GEO database.*

This transcriptional data and updated annotation files, alongside patient and region identifiers has been uploaded to the NCBI Gene Expression Omnibus (GEO) repository (<http://www.ncbi.nlm.nih.gov/geo/>) and is available under accession number GSE95109 and GPL23083. This data will be publically released on May 1st 2017.

- 7) *Text in many figures is too small, in particular the CMS-subtypes in Figure 2A, and any of the presented clusters.*

We have now included larger labels for all figures and increased the size of the colour indicators for patients/regions.

- 8) *Colors (like the grey and beige) or the CMS subtypes are not described in the legend of Figure 2, which makes the figure unclear.*

We have now included a colour label key and further description in the legends for all Figures that have used the Cayledo plots, to improve clarity of the Figures.

Minor:

The discussion starts with a repetition of what was said in the intro (line 221).

This has now been rectified.

What actually happens in the QC (that makes one of the data sets drop out) is unclear (line 315).

The QC parameters that resulted in these samples being excluded have now been outlined in the Methods section and Supplementary Figure 6.

Reviewer #2 (Remarks to the Author):

The manuscript by Dunne et al. addresses the challenge caused by intra patient heterogeneity in colorectal cancer to reliable classification by gene expression signatures. The authors have analysed RNA from 24 patients where they have measured gene expression by a CRC specific array from three samples per patient. That is, samples from centre of the primary tumour, its invasive front, and from a lymph node metastasis. Interestingly, and perhaps not surprisingly, signatures that have dealt with patient intrinsic expression produced more stable results across samples from the same patient. The extent of clustering based on patient-of-origin vs. region-of-origin was assessed for seven signatures, four clinically relevant (prognostic/predictive) signatures (Jorrenson, Eschrich, Kennedy, Popovici), two biologically subclassification signatures (CRIS, CMS) and a seventh signature, developed by the authors (30gene), as a positive control to be informative for the region-of-origin.

The results and conclusions of the papers are novel, interesting and important. Although I am not a statistician as such, I also find the types of analyses to be relevant and well documented. Since there are gene expression classifiers being proposed for several different types of cancer, such an evaluation of classifiers is of interest also to a wider (non-colorectal cancer researcher) audience. I liked reading this paper, and the text description of the results was convincing. However, I also found some major issues in the presentation of the results, which need to be addressed.

Major remarks:

The manuscript include many relevant analyses and plotted results from several gene expression profiles, but consistently shows results for only one or a few of these in various figures and tables. I understand that there may be space restrictions in the journal, but it would be a good reference to include similar analyses and plots for all seven signatures - at least as supplementary figures. Some (but not all) examples are the following:

- 1) *It is confusing that the CMS1 to 4 classifications are not included in many of the tabulations and figures where the four so-called clinically relevant signatures are evaluated and compared. E.g. why is the CMS1-4 and CRIS not included in e.g. Figures 1C+D and CMS in Figures 3 and 4, Supplementary Figures 2 and 3, Supplementary Table 2, and in the Methods description? I am of the opinion that this should be evident from the manuscript text, and not only in a rebuttal letter.*

We agree with Reviewer #2 that a comparison with CMS throughout the manuscript was necessary. Additionally, in line with point 1 from Reviewer 1, we provide a demonstration of the utility of the Sadanandam CRCA gene signature (Sadanandam et al Nat Med) as a surrogate marker for CMS (Supplementary Figure 1). As such, we can now include both an overall CMS assessment comparison for each subsequent analysis (Figure 1, 2, 3, 4, 5 and Supplementary Figure 1, 3, 4 and Supplementary Table 1 and 2). In addition, a more specific CMS4 (Stem-like) comparison which was the focus of our previous paper (Dunne et al CCR 2016) has been provided.

- 2) *Along the same line: CMS4 is scrutinised in Fig 1A, but why are similar plots not made for CMS1-3 and all the other signatures? At least such plots should be available as a Supplementary Figure. Without these plots as comparison/references, the Figure 1A seems useless as a proof of the intra-patient variation of CMS4 samples.*

Similar line plots are now included for CMS1, 2, 3 and 4 individually. The CMS classifier provides an enumeration of the random-forest (RF) probability of assignment to each CMS subgroup, thus allowing us to plot these values for comparative analysis. These line plots are not possible from the remaining "gene-list" signatures as, unlike the RF CMS classifier, there is no specific enumeration of assignment probability and as such we have used compared/contrasted all signatures using a variety of clustering and statistical methods. In addition, as described in our response to Reviewer #2 point 1 and Reviewer

#1 point 1, we now use the *Sadanandam* signature, and the focussed Stem-like signature, as surrogates of CMS and CMS4 respectively to allow comparison/reference to each signature throughout the study.

- 3) *Further, the Caleydo plot in Fig 2A is great to visualize the CMS intra-patient heterogeneity, but It would be much more informative by showing such a plot also for CRIS, which supposedly have better intra-patient concordance. And perhaps also include similar plots for the other signatures (in supplementary figs?)?*

The Caleydo plot directly comparing CMS and CRIS has now been presented in Figure 4D, which allows an assessment of the stability of these classification systems at different regions of a primary tumor. This is possible for the CMS random-forest classifier and the CRIS nearest template predictor (NTP) as the methodology involved in classification will result in a definitive assignment in each subtype. For the remaining prognostic/predictive signatures, as these will only result in assignment to 1 of 2 groups (e.g. good or bad, wildtype or mutant) we have presented multiple methods of clustering and statistical assessment to assess these further.

- 4) *In addition to the seven signatures evaluated here, there are also several published signatures which are not dealt with. The authors should better describe why they chose the particular signatures, and it would be interesting to know why other signatures were not chosen. Review articles have been published that go through most published/proposed signatures, which can serve as basis for the authors to list all signatures and why they did/didn't include them in the paper.*

We used the systematic review by Sanz-Pamplona and colleagues (PLoS One. 2012;7(11):e48877) as a basis for selecting the *Jorissen*, *Eschrich* and *Kennedy* signatures. The *Popovici* signature was developed from the PETACC-3 clinical trial cohort and was validated in additional clinical trial cohorts. These studies have all been very highly cited (demonstrating widespread use and consideration), have demonstrated prognostic utility and, in the case of the *Kennedy* signature, has received FDA approval. As such they were deemed to be clinically relevant and employed in this study. Further to this, the CMS and *Sadanandam* signature are now included as they represent the leading translational signatures currently in use. Following initial testing of the signatures (Figures 1, 2, 3), the findings indicated that cancer-cell intrinsic signatures might reduce the variation due to stromal ITH, therefore we validated this hypothesis using the CRIS signature from *Isella* and colleagues (Figure 4) followed by a cross comparison of all signatures (Figure 5). Reviewer #2 is correct in that there are many published signatures not dealt with in this study, but given the broad range of signatures and the variety of methodologies utilised, we feel that the eight different signatures chosen represent the most relevant/widely employed signatures and fully support the conclusions of this study.

- 5) *The authors write that the CRC disease specific array they used is produced by Almac. But more specifics should be included in the paper of what the content of this array is (e.g. number of probes and genes) in total and in relation to each of the described signatures. That is, the number and proportion of genes included for each of the signatures – as well as some of the principles behind how the genes in the almac were selected.*

These details have now been included in the Methods section and further detailed in Supplementary Table 1. To further answer Reviewer #2 point, we contacted Almac to obtain more specific details on this array and were provided with information from their patented technology:

The CRC disease specific array (DSA) produced by Almac Diagnostics Ltd. has a total of 61,528 probe sets and encodes 52,306 transcripts, which we have determined to represent 15,273 annotated genes. The CRC DSA provides robust profiling of RNA derived from FFPE tissue samples compared to Affymetrix microarrays, due to its disease specific content

(approximately 20,000 transcripts unique to the CRC DSA) and enhanced ability to detect degraded RNA by using a 3' based probe design.

These details are now outlined in the Methods section.

- 6) *The gene expression data was not available for me to review. I think this is fine, but the authors should describe how they will deposit the newly generated expression data (72 samples). The site (public repository) and accession number should be included in the manuscript.*

This transcriptional data and updated annotation files, alongside patient and region identifiers has been uploaded to the NCBI Gene Expression Omnibus (GEO) repository (<http://www.ncbi.nlm.nih.gov/geo/>) and is available under accession number GSE95109 and GPL23083. This data will be publically released on May 1st 2017.

Minor comments:

- 7) *P15. It is described that there were originally 25 patients, A to Y. Then that patient M was removed, so there were 24 patients left for further analyses. Then the name range is changed to A to X in p.5 line 100, although it is used A to Y in figures, leaving M out.*

This A to X typo has now been rectified in the text, the patient codes are indeed A to Y (without M).

- 8) *In Supplementary Table 1, cell E87, a gene symbol has been formatted as a date ("mar.03").*

This has now been rectified in Supplementary Table 1, and all annotations have been updated.

Reviewer #3 (Remarks to the Author):

In this study, Dunne et al build on their prior investigation of the impact of intra-tumor heterogeneity (ITH) in CRC on molecular subtyping (Clin Can Res 2016). Given the significant genomic, phenotypic and cellular ITH in diverse solid tumors, including CRC, this is an important issue. Indeed, previous studies have demonstrated that different regions of the same tumor can be classified into distinct molecular subtypes, posing challenges for precision medicine. In addition to genetic heterogeneity, immune cell infiltration and stromal cells contribute to apparent ITH resulting in a reduction in the purity of the tumor cell population and, signal attenuation in various molecular assays. However, it is increasingly appreciated that stromal/immune cell infiltration is an inherent feature of the tumor microenvironment and enriched in specific molecular subgroups.

Here, the authors extend their previous analysis of multi-region expression data for 24 primary CRCs (central tumor and invasive front) and matched lymph node mets on a custom CRC array. They report that the gene expression data cluster by patient rather than tissue of origin. They further note that cancer-cell intrinsic signatures tended to result in more robust patient-specific classifications than other signatures, thereby overcoming the issue of ITH.

While the manuscript addresses an interesting topic, it is not clear that this represents a significant advance over the authors previous work on this topic. I also have concerns regarding aspects of the methods, and therefore the claims that are asserted.

Specific comments:

- 1) *Much of the manuscript is focused on the evaluation sample clustering using different signatures. However, given the extensive variation in gene expression across patients, it does not seem surprising that samples from the same patient tend to cluster more tightly than samples from different patients of similar histology/morphology even in the face of differences in stromal composition. Moreover, most gene expression arrays include probes with SNPs that can result in SNP-expression associations. While it is not clear as to how many such probes exist on the custom array employed here (details on this are scant), this could also contribute to "within patient" signals.*

When all transcripts on a large transcriptomic array are being examined, tumour area is unlikely to have a large effect on the majority of transcripts on the array. Unsupervised clustering of all gene expression probesets do generally cluster tightly according to patient-of-origin, even when the tissue is derived from different regions of the primary tumor architecture. As Reviewer #3 points out, clustering of patients is likely due to both transcription levels and SNP-expression associations. Despite this global transcriptomic similarity according to patient of origin, the interpretation of transcription-based prognostic signatures are ultimately dependent on the gene expression profiles of small pre-defined lists of individual genes. When classification of samples is distilled to this level of granularity, the subtle changes observed at different regions may result in misclassification, similar to the issues we reported for the CMS classifier due to stromal-derived genes in our Clinical Cancer Research paper. This is most relevant in prospective stratification of patients using colonoscopy or biopsy tissue, where the region-of-origin of the sample from within the three-dimensional tumor architecture is unknown. The testing of the ability of clinically informative signatures in the face of this potentially confounding multi-regional sampling has been performed in this study. As the results show, certain classifiers are undermined by multi-regional sampling, while signatures such as the Popovici and CRIS (primarily based on epithelial genes) limit this confounding issue.

To fully address this point from Reviewer #3, we again contacted Almac directly in regard to the proportion of SNP-probes on the DSA array. We were informed that given the array design, from deep sequencing of CRC-specific transcripts, in addition to where probes are confined, mainly the 200bp at the 3' end of transcripts to ensure stability in FFPE tissue, that it is highly unlikely that SNP-associated transcriptional changes could have affected our study. Almac do offer a specific SNP

array, and as such these SNP regions would have been avoided in the DSA array design.

- 2) The authors also evaluate the classification of LN mets relative to the primary. While mets may harbor a different stromal signature, they are nonetheless expected to harbor similar expression profiles to the primary (as has been shown in breast cancer). Hence, the finding that subtypes are generally similar between primary and LN met is not particularly surprising.

Similar to the discussion above in Reviewer #3 Response Point 1, our findings have demonstrated that when using small defined lists of genes, there is a very broad range in the ability of some signatures to concordantly subtype samples from the primary tumour and lymph nodes. Using signatures that are more dependent on epithelial genes, and therefore less dependent on stroma, appear to perform best from our analysis. As Reviewer #3 indicates, this finding is intuitive, but it had not been fully tested before using subtyping or prognostic/predictive signatures in colorectal cancer. We have included an additional section focussed on the issues raised by Reviewer #3 Response Points 1 and 2 within the Discussion section.

- 3) *In terms of signature development, there is no discussion as to whether all genes included in the original signatures are represented on the custom array employed here, . If there was gene dropout (as seems to be the case based on Table S1), this must be noted and the impact of this information loss on the performance of the published signature evaluated (for example with existing public data).*

We agree with Reviewer #3 that the impact of gene “drop-out” may become an issue when testing published gene signatures derived from a variety of different methodologies. The platforms used in the generation of the gene signatures used in this study include Affymetrix and custom cDNA arrays alongside NGS technology. Inevitably, when comparing the utility of these signatures, there will be some cases when individual genes/probes are not universally represented. In order to minimise this dropout, we initially used core-genes from each signature, which were derived using the methodology employed by Sanz-Pamplona and colleagues (PLoS One, 2012;7(11):e48877), a method previously developed to allow comparison and testing of gene signatures from different platforms.

In response to Reviewer #3's point, we have again taken steps to further minimise any gene dropout and have revisited the array annotation file and updated any missing gene symbols, if they have recently been annotated. Additionally, we have used a combination of Entrez ID, Unigene, HUGO and Refseq to comprehensively update the annotation file for the DSA array. Despite these efforts, inevitably no match was found for a small number of genes in some signatures and they were lost for subsequent analysis. The levels of gene drop out were minimal, with 0%, 10.3%, 2.9%, 2.3%, 3.5%, 0% and 8.0% of genes being lost for the 30 gene, *Eschrich*, *Jorissen*, *Sadanandam*, *Kennedy*, *Popovici* and CRIS signature respectively. The signature with the highest dropout (*Eschrich* 10.3%) was because of only three genes missing, from a core gene signature of 29.

To further test the implications of drop-out, we performed a head-to-head comparison of the *Kennedy* signature using the translated core genes (which can be tested on both DSA and non-DSA arrays) compared to the probset IDs defined in the original publication (which can only be tested on a DSA array). These additional probesets in the full *Kennedy* signature had no gene annotation and were derived from either clone-based sequences, non-coding RNA or had no transcript/genomic match and were therefore unique to this array.

This comparison between the “dropout” core gene signature and the original signature indicated only a minor change in specific analyses values, but no change in the overall findings. The use of the probe-specific list results in a slight reduction in performance at initial subgrouping (2 clusters) and a slight increase in individual patient clustering (24 clusters) when compared to the signature with the gene dropout (data shown below).

As it would not be possible to perform the same analysis for all other signatures (as our array did not have probes representing all genes in all signature) we proceeded with the core genes methods defined by Sanz-Pamplona and colleagues ([PLoS One](https://doi.org/10.1371/journal.pone.0188877). 2012;7(11):e48877) in order to make our study more reproducible using different transcriptomics technologies/platforms. The inherent limitation of gene signature testing and validation due to non-overlapping gene annotation has been further stressed in the Methods and Discussion sections. Our comprehensively updated annotation file has also been uploaded to the GEO public repository.

Given this limitation, the signatures used in our study can only serve as a representation of the original signatures. Importantly however, our study is focussed on dissecting the cellular source of the core genes related to their ability to robustly cluster patient samples to outline parameters that could potentially improve future signature/classifier design. This has been further outlined in the Methods section.

- 4) *Several CRC signatures are evaluated in this manuscript including a CRC intrinsic signatures (CRIS). However, there is no further comparison with the consensus molecular subtypes (as was done in the Dunne 2016 paper). This should be included as additional comparisons are presumably made here (for example comparison of primary tumor regions and matched lymph node metastases).*

We agree with Reviewer #3 that a comparison with CMS throughout the manuscript is necessary. Additionally, in line with point 1 from Reviewer #1 and point 2 from Reviewer #2, we have now included more detailed analyses with CMS (Figure 1) and the CMS surrogate signature from *Sadanandam* throughout (Figure 1, 2, 3, 4, 5 and Supplementary Figure 1, 3, 4 and Supplementary Table 1 and 2).

- 5) *The authors further assert that the Popovici and CRIS signatures have clinical utility since they retain patient-specific clustering. However, this does not equate to clinical utility. Rather, this would necessitate that the signatures provide additional prognostic value beyond established clinical covariates. This is not assessed in this manuscript, and the claim must therefore be toned down.*

Our description of clinical utility has been further defined and the claim has been tempered to reflect this point.

6) *The text emphasizes that CRIS enables concordant patient sample clustering. However, this is also true for the Popovici signature; therefore the discussion of the relative merits should be more balanced.*

The *Popovici* signature has been included as having an equal ability to cluster in the Conclusion section to reflect Reviewer #3's point.

To undertake the additional analysis to address the Reviewers points, as outlined in this rebuttal, we have made 2 changes to the authorship on our manuscript; namely Matthew Alderdice has now moved to 2nd author and Amy M.B. McCorry has been added as a co-author. These changes have been approved by all co-authors.

Again, we would like to thank the Editor, Reviewers and editorial team for the very positive comments and the opportunity to improve our manuscript.

Regards,
Dr Philip Dunne and Prof. Mark Lawler.

Reviewers' Comments:

Reviewer #1:

Remarks to the Author:

The manuscript has substantially improved by introductions of the comparison to CMS subtypes in text and figures. This has brought it to a level that warrants publication in Nat Comm. Our major concerns have been addressed.

We regret the authors did not elaborate in their reply to our comments what changes were made in the manuscript text. For example "the added value of the 2016 paper" was not included in their reply letter

Reviewer #2:

Remarks to the Author:

I find that the authors have replied convincingly to my original criticism and made satisfactory revisions.

Reviewer #3:

Remarks to the Author:

The authors have satisfactorily addressed the concerns raised in the previous round of review.